

# Associations between physical fitness, body composition, and heart rate variability during exercise in older people: exploring mediating factors

Diego Mabe-Castro[1,2], Matías Castillo-Aguilar[1], Matías Mabe-Castro[1,3], Ruby Méndez Muñoz[1], Carla Basualto-Alarcón[4,5] and Cristian Andrés Nuñez-Espinosa[1,3,6]

[1] Centro Asistencial Docente e Investigación, Universidad de Magallanes, Punta Arenas, Chile
[2] Departamento de Kinesiología, Universidad de Magallanes, Punta Arenas, Chile
[3] Escuela de Medicina, Universidad de Magallanes, Punta Arenas, Chile
[4] Health Sciences Department, University of Aysén, Coyhaique, Chile
[5] Anatomy and Legal Medicine Department, Universidad de Chile, Santiago, Chile
[6] Interuniversity Center for Healthy Aging, Chile, Chile

Corresponding author
Cristian Andrés Nuñez-Espinosa, cristian.nunez@umag.cl

## ABSTRACT

**Background**. Age-related changes in body composition affect physical fitness in older adults. However, whether the autonomic response is associated with body fat percentage and its implication for physical fitness is not fully understood.

**Aim**. To understand the association between physical fitness, body composition, and heart rate variability in older people and its mediating factors.

**Methods**. A cross-sectional study with 81 older adults was conducted, assessing Short Physical Performance Battery (SPPB), Two-minute Step Test (TMST), body composition, and cardiac autonomic response. Correlation and mediation analyses were performed.

**Results**. Body fat percentage negatively correlated with physical fitness (SPPB: $r = -0.273$, $p = 0.015$; TMST: $r = -0.279$, $p = 0.013$) and sympathetic activity (sympathetic nervous system (SNS) index: $r = -0.252$, $p = 0.030$), yet positively correlated with parasympathetic tone (root mean square of successive differences (RMSSD): $r = 0.253$, $p = 0.029$; standard deviation of NN intervals (SDNN): $r = 0.269$, $p = 0.020$). Physical fitness associated with sympathetic nervous system index (SPPB: $r = 0.313$, $p = 0.006$; TMST: $r = 0.265$, $p = 0.022$) and parasympathetic nervous system index (TMST: $r = -0.344$, $p = 0.003$). Muscle mass mediated body fat's impact on physical fitness, while physical fitness mediated body fat's impact on autonomic response.

**Conclusion**. Body composition and cardiac autonomic response to exercise are associated with physical fitness in older people, highlighting a possible protective effect of muscle mass against the decline in physical fitness associated with increased body fat.

## INTRODUCTION

Aging is accompanied by a multitude of physiological changes that have the potential to significantly impact people's overall health and well-being (*Partridge, Deelen & Slagboom, 2018*; *Feng, 2019*). Body composition and cardiovascular function are particularly relevant, as they are closely associated with aging (*Malandrino et al., 2023*; *Liu et al., 2023*; *Ferrucci & Fabbri, 2018*; *Xie et al., 2023*; *Gielen et al., 2021*). Recently, there has been a growing interest in comprehending the interplay between physical fitness, body composition, and cardiac autonomic response to exercise in older individuals, as these factors play pivotal roles in determining health outcomes within this age group.

Physical fitness is the ability to perform daily tasks with vigor, without undue fatigue, and with ample energy to enjoy leisure-time pursuits and meet unforeseen emergencies (*Siscovick, LaPorte & Newman, 1985*). Therefore, it is a crucial component of healthy aging. The term encompasses many aspects, including cardiorespiratory endurance, muscular endurance and strength, body composition, and flexibility (*Caspersen, Powell & Christenson, 1985*). Numerous studies have demonstrated the positive impact of physical fitness on overall health and longevity, emphasizing its role in reducing the risk of chronic diseases and improving quality of life and functional independence in older adults (*Lee, Paffenbarger Jr & Hennekens, 1997*; *Strasser & Burtscher, 2018*; *Yang et al., 2019*; *López-Bueno et al., 2022*).

Moreover, the alteration of body composition, specifically the proportion of body fat, is accompanied by significant changes with age, resulting in a tendency towards an increase in adiposity and a decrease in lean muscle mass (*Palmer & Jensen, 2022*; *Pataky, Young & Nair, 2021*). Excessive body fat accumulation, particularly visceral adiposity, has been associated with a higher risk of cardiovascular disease, metabolic disorders, and functional limitations in older individuals (*Neeland et al., 2019*; *Powell-Wiley et al., 2021*).

The autonomic nervous system (ANS) plays a critical role in regulating cardiovascular function, with sympathetic and parasympathetic branches exerting opposing effects on heart rate and vascular tone. Furthermore, during stressful situations, such as physical exercise, the ANS ensures a sufficient cardiac response to higher metabolic demands (*Freeman et al., 2006*). Heart rate variability (HRV) is a non-invasive indicator of ANS activity and cardiovascular health (*Zhao et al., 2024*; *Kubota et al., 2017*), usually measured by electrocardiogram or by a wireless heart rate sensor installed *via* a chest strap. It encompasses a wide range of variables derived from mathematical calculations based on heart rate records. Therefore, the variables used in this study are defined in the Instruments section. A reduction in resting HRV has been linked to various adverse outcomes, including cardiovascular events and morbidity, while insufficient HRV reduction during exercise may lead to impaired physical capacity (*Kubota et al., 2017*; *Tiwari et al., 2021*; *Mongin et al., 2022*).

However, HRV is not only influenced by physical and environmental stressors. Psychological factors such as anxiety and depression have also been linked to changes in ANS activity (*Brown et al., 2018*; *Cheng et al., 2022*). Older individuals may be more

susceptible to these psychological factors, making it crucial to measure and control for their effects (*Schlechter, Ford & Neufeld, 2023*; *Andreescu & Varon, 2015*).

Despite extensive research into physical fitness, body composition, and cardiovascular health in older adults, gaps persist in understanding their interrelationships and the factors mediating them. One area of interest is the relationship between physical fitness, body fat percentage, and cardiac autonomic response to exercise in older individuals. This study explores their collective influence on physical fitness in the aging population.

The conceptualization of this investigation is based on recognizing the interplay between physical fitness, body composition, and autonomic cardiovascular regulation in aging. We aim to answer the following research question: How does physical fitness relate to body fat percentage and cardiac autonomic responses to exercise in older people? By elucidating these correlations, we hope to gain valuable insights into the relationships that underlie age-related modifications in physical fitness and identify potential interventions to enhance health outcomes in older adults. Even more, we seek to understand the mediating effect of the variables, including physical aspects and frequent psychological traits, such as geriatric depression and anxiety.

We hypothesized that physical fitness would be inversely related to body fat percentage and associated with cardiac autonomic response during exercise in older individuals. We believe an optimal cardiac autonomic response, characterized by a shift towards sympathetic activity predominance, will be essential for this population to achieve greater performance during physical fitness testing.

## MATERIALS & METHODS

### Aims and study design

A prospective observational, cross-sectional study was conducted in one data collection session to determine the associations between physical fitness, body composition, and cardiac autonomic response to moderate-intensity exercise.

### Setting

This study was conducted at the Centro Asistencial Docente e Investigación (Assistance, Teaching, and Research Center), which belongs to the University of Magallanes (CADI-UMAG) in Punta Arenas, Chile. All the assessments were made between 09:00 and 11:00 a.m. The privacy and comfort of the subjects were ensured; the room temperature was set at 20°C, and white artificial lighting was used.

### Participants

A total of 81 community-dwelling older adults were recruited and selected by non-probabilistic sampling from the Region of Magallanes and Chilean Antarctica, Chile. If they were 60 years or older, residents of the Region of Magallanes and Chilean Antarctica, Chile, they were included and understood the study aims and instructions. However, they were excluded in the case of diagnosis of congenital heart disease, consumption of beta-blocker drugs, taking stimulant substances within 24 h before assessment session, motor or cognitive disability, inability to understand instructions or written content, or presence of pain during cardiac or physical assessments.

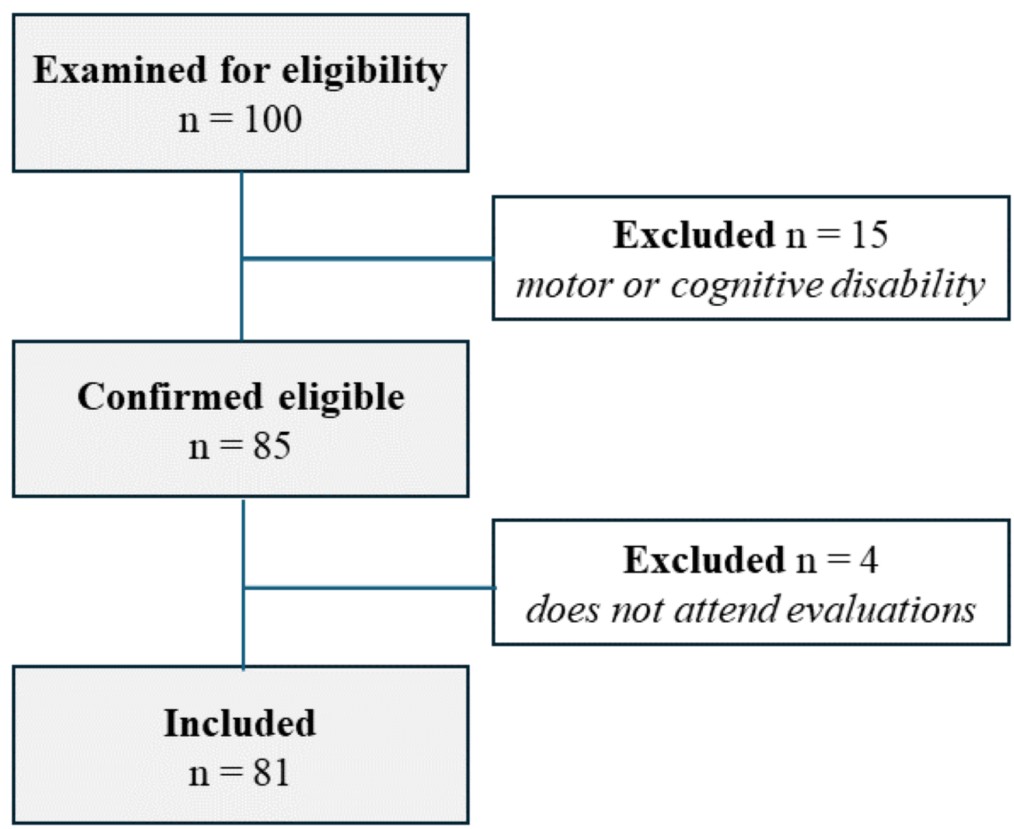

**Figure 1** **The flow diagram of the included participants.**

All participants gave their permission and provided informed written consent before participation. The Ethics Committee of the University of Magallanes (N° 10/CEC-UMAG/2023) approved this study, following the regulations established by the Declaration of Helsinki on ethical principles in human beings.

The flow diagram of the included participants can be seen in Fig. 1.

## Procedures

During recruitment, the participants were instructed to avoid the use of psychoactive substances for 24 hours before the assessment and to sleep for at least 7 hours the previous night. Upon arrival, participants were informed about the study aims and risks associated with their participation, and sociodemographic and medical information was collected during the initial interview. Then, body composition parameters were measured using bioimpedance analysis, and psychological questionnaires for anxiety and depression screening were applied by a supervised psychology undergraduate student.

Cardiac autonomic response to physical exercise was measured as previously validated for this age group (*Castillo-Aguilar et al., 2023*). The protocol is briefly described below:

HRV was recorded through a non-invasive chest band immediately before, during, and after executing the two-minute step test (TMST). Vital signs, including blood pressure,

were monitored throughout the test, and participants' well-being was visually checked to ensure they were comfortable and prepared. For resting HRV measurements (before and after the application of the TMST), the volunteers remained seated in a chair, with feet and back supported, ensuring avoidance of talking during the recordings. R-R intervals were recorded continuously during the last 10 min of rest and were analyzed for 5 min on each occasion. The breathing rate was spontaneous. As part of the protocol, it was ensured that the participant had a blood pressure of less than 140/90 mmHg to start the HRV measurements.

Finally, after 15 minutes of resting from the TMST, the Short Physical Performance Battery (SPPB) was administered to the participants. Before the session ended, the subjects' blood pressure, heart rate, and general appearance were verified.

Physical and physiological assessments, as well as the initial interview, were made by professional physiotherapists.

## Instruments
### Short physical performance battery
SPPB consists of a physical test used to measure three components of physical fitness, described below (*Guralnik et al., 1994*):

Balance: to achieve the maximal score (4 points), the subject should be able to stay balanced for at least 10 seconds in a side-by-side stand (feet together), semi-tandem stand, and tandem stand.

Usual gait speed: The subjects are asked to walk normally at a 4-meter distance. They are given two chances, and the best of both is registered. To achieve the maximal score (4 points), they should obtain 4.82 sec. or less.

Lower body fitness: The subjects are asked to perform five chair stands without using their arms, and time is registered upon completion. A time lower or equal to 11.19 sec. gives the maximal score (4 points).

To obtain the final score, the sum of the three components is calculated, with a maximal total score of 12 points.

### Body composition
Body mass (kg) and total body fat (%) were assessed by bioimpedance using the Tanita BC-558 Ironman Segmental Body Composition Monitor (Tanita Ironman, Arlington Heights, IL, USA), with a concordance of 89.3% compared to the Dual X-ray Absorption test using standard measurement protocols (*Mialich, Martinez & Júnior, 2011*).

### Cardiac autonomic activity
The cardiac autonomic activity was assessed using R-R interval recordings (HRV) obtained through the Polar Team2 system (Polar®) application. Artifacts and ectopic heartbeats were excluded, not exceeding 3% of the recorded data (*Malik, 1996*). Time-domain parameters considered for analysis included the square root of the mean squared differences of successive R-R intervals (RMSSD, expressed in ms) as an index of parasympathetic activity (*Buchheit et al., 2010*) and the standard deviation of RR intervals (SDNN), reflecting total variability encompassing both sympathetic and parasympathetic contributions to

cardiac autonomic function (*Berntson et al., 1997*; *Buchheit & Gindre, 2006*). The Stress Index (SI) and Parasympathetic and Sympathetic Nervous System Index (PNS and SNS) were computed. The PNS Index, indicative of total vagal stimulation, was derived from mean R-R intervals, RMSSD, and Poincaré Plot Index SD1 in normalized units (linked to RMSSD), representing deviations from normal population averages (*Berntson et al., 1997*; *Rajendra Acharya et al., 2006*). The SNS Index, reflecting total sympathetic stimulation, was derived from mean R-R intervals, Baevsky's Stress Index (positively related to cardiovascular system stress and cardiac sympathetic activity), and the Poincaré Plot Index SD2 in normalized units (related to SDNN) with interpretation similar to the PNS Index (*Berntson et al., 1997*; *Rajendra Acharya et al., 2006*). The SI indicates the ANS control system's workload (*Yoo et al., 2020*), normalized by the square root of Baevsky's SI (*Baevsky, 2008*). All analyses were conducted to compute HRV-related indices using Kubios HRV® software (Kuopio, Finland).

### Two-minutes step test

The TMST is a sub-test from the Senior Fitness Test, demanding a low to moderate intensity (*Castillo-Aguilar et al., 2023*; *Rikli & Jones, 1999*). It consists of a two-minute assessment designed to evaluate cardiorespiratory fitness. Participants are instructed to raise their right knee to a marked point on a wall as often as possible within the given time frame, ensuring that each raise reaches at least a 70° angle at the thigh-femoral joint. The number of valid steps was recorded for each subject.

### Geriatric depression scale

The 30-question Geriatric Depression Scale (GDS-30) was employed to assess the subject's depressive symptoms (*Gana et al., 2017*). It consists of a dichotomous questionnaire, where participants are asked about their past-week feelings about depressive symptoms (for instance, "Do you feel that your life is empty?"), with higher scores (*i.e.,* "yes" responses) representing more depressive symptoms. It was first developed by *Yesavage et al. (1982)* and is currently widely used (*Zenebe et al., 2021*). The Spanish version employed in this study has been previously validated, with a Cronbach alpha coefficient of 0.82 (*Fernández-San Martín et al., 2002*). The authors have permission to use this instrument from the copyright holders.

### Beck anxiety inventory

The Beck Anxiety Inventory was employed to assess anxiety symptoms. It consists of a 21-item questionnaire based on usual anxiety symptoms and a 4-option Likert scale from 0 ("Not at all") to 3 ("Severely"), meaning the severity reported by the subject in each one. It was originally developed by *Beck et al. (1988)*. This study uses the Spanish version, demonstrating a high internal consistency in older people ($\alpha = 0.94$) (*Rodríguez Reynaldo, Martínez Lugo & Rodríguez Gómez, 2001*). The authors have permission to use this instrument from the copyright holders.

## Statistical analysis

We used mean and standard deviation (SD) to describe continuous variables and absolute and relative frequencies to describe discrete variables. We used Pearson's product-moment

correlation (r) to assess the relationship between continuous variables and Spearman's rank correlation (rho) to evaluate the relationship between body composition variables and the ordinal constituents of the SPPB. To assess differences between groups, we used standardized mean difference (SMD) and 95% confidence intervals (CI$_{95\%}$).

As a way of controlling for the influence of psycho-physiological variables on the cardiac autonomic response of the subjects to exercise, we assess the average causal mediation effect (ACME), the average direct effect (ADE) of the main effects after taking into account the effect of moderator variables into the observed relationships and the proportion of the effect that its mediated by these variables (*Imai, Keele & Yamamoto, 2010*). To assess the significance of moderating variables, we used nonoverlapping CI$_{95\%}$, estimated based on nonparametric bootstrapping using Monte Carlo resampling, and bias-corrected. We accelerated CI$_{95\%}$, using the *mediation* R package to this end (*Tingley et al., 2014*).

We defined a type I error rate of 5% ($\alpha = 0.05$) as our threshold for null hypothesis significance testing and nonoverlapping CI$_{95\%}$ over the null effect for sex differences and mediation analyses.

To estimate the confidence in our conclusions and, consequently, the statistical power of our analyses, we calculated the probability of correctly rejecting the null hypothesis for the correlation tests post hoc, considering a low (*i.e.*, $r = 0.3$) to moderate effect size ($r = 0.5$).

All analyses were computed using the *R* programming language for statistical computing (*R Core Team, 2021*).

## RESULTS

### Sample characteristics

The sample consisted of 81 adults, averaging 71.1 years ($\pm 6.2$), with a BMI of 31 ($\pm 6.2$ kg/m$^2$), and predominantly female (82.7%, $n = 67$). Table 1 presents the detailed characteristics of the sample.

### Body composition and autonomic response

We observed a negative correlation between body fat percentage and the sympathetic nervous system (SNS) response ($r = -0.252$, $p = 0.030$) and the Stress Index ($r = -0.258$, $p = 0.027$) during the TMST. Conversely, body fat percentage positively correlated with heart rate variability (HRV) metrics during exercise, specifically RMSSD ($r = 0.253$, $p = 0.029$) and SDNN ($r = 0.269$, $p = 0.020$) (See Fig. 2).

When evaluating the sex-specific effects on autonomic response, most correlations held true for females. Body fat percentage correlated negatively with the SNS Index (females: $r = -0.356$, $p = 0.005$; males: $r = 0.174$, $p = 0.589$) and the Stress Index (females: $r = -0.316$, $p = 0.012$; males: $r = 0.204$, $p = 0.525$). However, there was no sufficient statistical evidence to suggest a sex-specific response for RMSSD (females: $r = 0.246$, $p = 0.054$; males: $r = -0.259$, $p = 0.416$) and SDNN (females: $r = 0.247$, $p = 0.053$; males: $r = -0.058$, $p = 0.859$). No other body composition variables showed a significant correlation with cardiac autonomic response to exercise.

**Table 1 Main sample characteristics.** Body composition and characteristics are displayed for the overall sample and separated by sex. Differences between males and females are displayed as SMD and CI[95%] for continuous variables.

| Characteristic | Overall, N = 81[1] | Sex | | Difference[2] | 95% CI[23] |
| --- | --- | --- | --- | --- | --- |
| | | Female, N = 67[1] | Male, N = 14[1] | | |
| Age (years) | 71.5 ± 5.6 | 71.4 ± 5.7 | 71.9 ± 5.5 | −0.08 | −0.66, 0.50 |
| Hypertension | 33 (41%) | 26 (39%) | 7 (50%) | | |
| Diabetes | 12 (15%) | 9 (13%) | 3 (21%) | | |
| Body weight (kg) | 74.6 ± 13.2 | 72.8 ± 12.6 | 85.0 ± 12.7 | −0.98 | −1.6, −0.33[*] |
| Height (cm) | 155.8 ± 9.2 | 153.3 ± 7.6 | 167.8 ± 6.3 | −2.1 | −2.8, −1.4[*] |
| BMI (kg/m2) | 31.0 ± 6.2 | 31.2 ± 6.5 | 29.7 ± 3.2 | 0.28 | −0.36, 0.92 |
| BMI category | | | | | |
|    Normal | 7 (9%) | 6 (9%) | 1 (9%) | | |
|    Overweight | 35 (45%) | 29 (43%) | 6 (55%) | | |
|    Obese | 36 (46%) | 32 (48%) | 4 (36%) | | |
| Muscle mass (kg) | 43.9 ± 8.2 | 41.0 ± 4.3 | 59.9 ± 6.5 | −3.5 | −4.4, −2.7[*] |
| Fat mass (%) | 37.5 ± 9.3 | 39.5 ± 8.3 | 26.2 ± 6.4 | 1.8 | 1.2, 2.5[*] |
| Bone mass (%) | 2.3 ± 0.4 | 2.2 ± 0.2 | 3.1 ± 0.3 | −3.6 | −4.5, −2.8[*] |
| Water (%) | 47.0 ± 6.7 | 45.4 ± 5.6 | 55.8 ± 5.0 | −2.0 | −2.7, −1.3[*] |

Notes.
[*] shows significant sex differences.

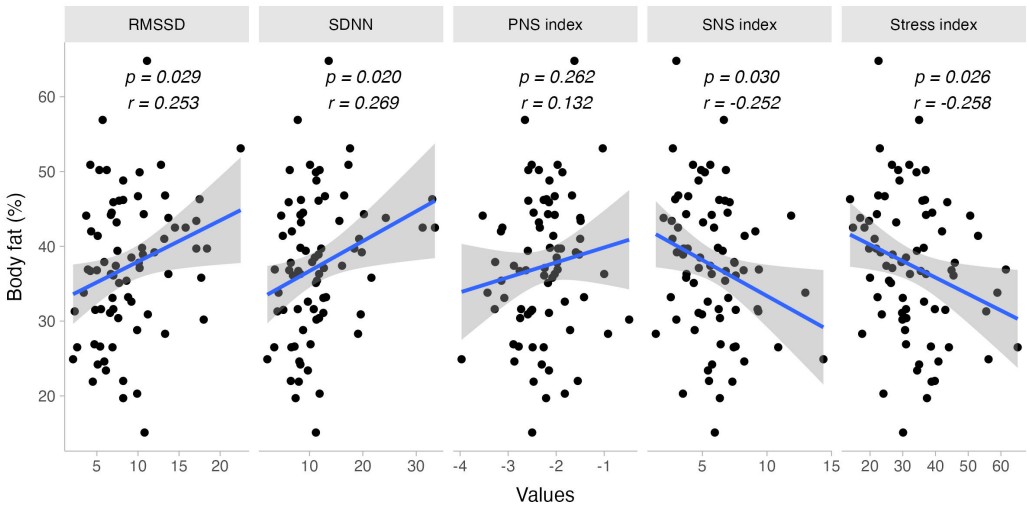

**Figure 2 Bivariate dispersion plots between body fat and HRV-related measures.** Significance values for Pearson's product-moment correlation test are shown.

## Body composition and physical fitness

Lower body fat percentage was associated with higher SPPB scores in the sit-to-stand (rho = −0.279, p = 0.013) and gait speed (rho = −0.261, p = 0.021) subtests, but not in the balance score (rho = 0.052, p = 0.656). Similarly, lower BMI values correlated with higher SPPB scores in sit-to-stand (rho = −0.325, p = 0.004) and gait speed (rho = −0.305, p = 0.007), but not in balance (rho = −0.083, p = 0.474).

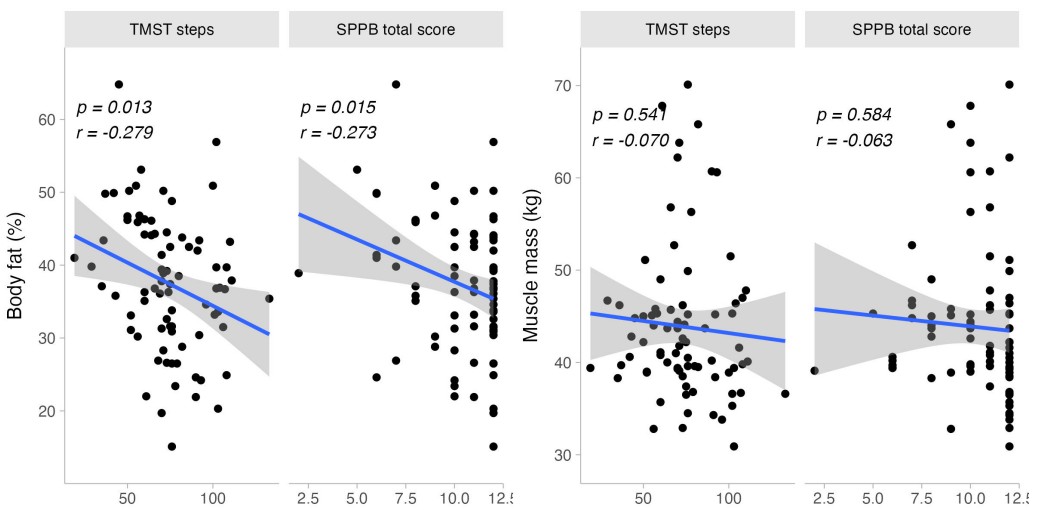

**Figure 3  Bivariate dispersion plots between physical fitness and body composition-related measures.** Significance values for Pearson's product-moment correlation test are shown.

Furthermore, total steps from the TMST were inversely correlated with body fat percentage (r = −0.279, p = 0.013) and body weight (r = −0.232, p = 0.041). No significant correlations were found between total muscle mass and SPPB scores (r = −0.063, p = 0.584) or TMST steps (r = −0.070, p = 0.541) (See Fig. 3).

Sex-specific analyses revealed relationships between body composition variables and physical fitness measures. For females, body fat percentage and BMI were linked to sit-to-stand (body fat: rho = −0.347, p = 0.004; BMI: rho = −0.324, p = 0.008) and gait speed (body fat: rho = −0.309, p = 0.012; BMI: rho = −0.318, p = 0.009). However, these correlations were not significant in males (sit-to-stand, body fat: rho = −0.304, p = 0.337; BMI: rho = −0.294, p = 0.380; gait speed, body fat: rho = 0.172, p = 0.593; BMI: rho = −0.108, p = 0.752).

Identical findings were observed for TMST total steps, with significant correlations for females between steps and body fat percentage (r = −0.311, p = 0.01) and body weight (r = −0.275, p = 0.024). No such correlations were found in males (steps, body fat: r = 0.085, p = 0.792; body weight: r = −0.198, p = 0.559).

## Physical fitness and cardiac autonomic response to exercise

Physical fitness, reflected through total steps in TMST and SPPB sit-to-stand score, was positively correlated with greater sympathetic activity during exercise. This was indicated by the SNS Index (SPPB sit-to-stand: rho = 0.345, p = 0.003; TMST steps: r = 0.265, p = 0.022), mean HR (SPPB sit-to-stand: rho = 0.387, p = 0.001; TMST steps: r = 0.338, p = 0.003), and its counterpart mean R-R interval (SPPB sit-to-stand: rho = −0.394, p = 0.001; TMST steps: r = −0.311, p = 0.007). Other SPPB sub-domains did not exhibit this behavior (*i.e.*, p >0.05 for balance and gait speed scores).

Sex-specific analyses showed that the relationship between the SNS Index during exercise and the SPPB sit-to-stand score was stronger in males (rho = 0.607, p = 0.028) compared

to females (rho = 0.294, p = 0.021). However, the correlation between total TMST steps and the SNS Index was significant only in females (r = 0.326, p = 0.01) and not in males (r = −0.211, p = 0.51). A similar sex-specific effect was observed between TMST steps and mean HR during exercise (females: r = 0.414, p = 0.001; males: r = −0.17, p = 0.598) and mean R-R interval (females: r = −0.398, p = 0.001; males: r = 0.16, p = 0.62).

The total number of TMST steps was inversely correlated with parasympathetic indices during exercise, such as RMSSD (r = −0.285, p = 0.014) and PNS Index (r = −0.344, p = 0.003). Sex-specific analyses indicated similar patterns for RMSSD (females: r = −0.288, p = 0.023; males: r = −0.08, p = 0.805) and PNS Index during exercise (females: r = −0.41, p = 0.001; males: r = 0.115, p = 0.722).

Additionally, there was a proportional decrease in the PNS Index during exercise with increasing levels of physical fitness, as shown by the SPPB sit-to-stand score (rho = −0.407, p <0.001). A similar effect was observed for the PNS Index post-exercise, particularly in the SPPB balance score for females (females: rho = −0.259, p = 0.046; males: rho = −0.077, p = 0.802), unlike the sit-to-stand score, which showed a negative correlation in males only (females: rho = −0.087, p = 0.506; males: rho = −0.570, p = 0.042).

Likewise, RMSSD during exercise decreased with increasing physical fitness, as indicated by the sit-to-stand score (rho = −0.327, p = 0.005). This relationship was significant only for females (females: rho = −0.333, p = 0.009; males: rho = −0.419, p = 0.154) (See Fig. 4).

## Mediation analysis

Mediation analyses suggest many potential influential effects on the relationships between body composition and parasympathetic indices. In this context, the average direct effect (ADE) of body fat percentage on RMSSD during exercise (ADE = 0.088, $CI_{95\%}$ [−0.005, 0.200], $p = 0.077$) was accentuated when considering the influence of SPPB score (ACME = 0.0289, $CI_{95\%}$ [0.003, 0.110], $p = 0.057$; Total effect = 0.117, $CI_{95\%}$ [0.028, 0.240], $p = 0.012$). Similar findings were observed when considering the mediation effect of TMST steps into account (ACME = 0.0283, $CI_{95\%}$ [0, 0.07], $p = 0.079$; Total effect = 0.117, $CI_{95\%}$ [0.028, 0.240], $p = 0.012$). No other parasympathetic indicators were influenced or mediated by fitness or psychological-related measures.

Additionally, and in the case of sympathetic indicators, the effect of body fat on SNS index during exercise (ADE = −0.050, $CI_{95\%}$ [−0.107, 0.01], $p = 0.077$) was influenced by SPPB score in similar way as with RMSSD, enhancing the original main observed effect (ACME = −0.016, $CI_{95\%}$ [−0.049, 0], $p = 0.041$; Total effect = −0.066, $CI_{95\%}$ [−0.125, −0.010], $p = 0.016$). Similar effects were observed with TMST steps in this regard (ACME = −0.015, $CI_{95\%}$ [−0.042, 0], $p = 0.050$; Total effect = −0.066, $CI_{95\%}$ [−0.125, −0.010], $p = 0.016$). No other sympathetic indices were notoriously modified in the presence of either SPPB score, TMST steps, or psychological variables.

When assessing potential mediators for the effect of body composition on fitness-related measures, we found that the impact of body fat percentage on SPPB score (ADE = −0.075, $CI_{95\%}$ [−0.123, −0.020], $p = 0.007$) is partially diminished when considering the influence of total muscle mass (ACME = 0.011, $CI_{95\%}$ [0, 0.040], $p = 0.107$; Total effect = −0.064,

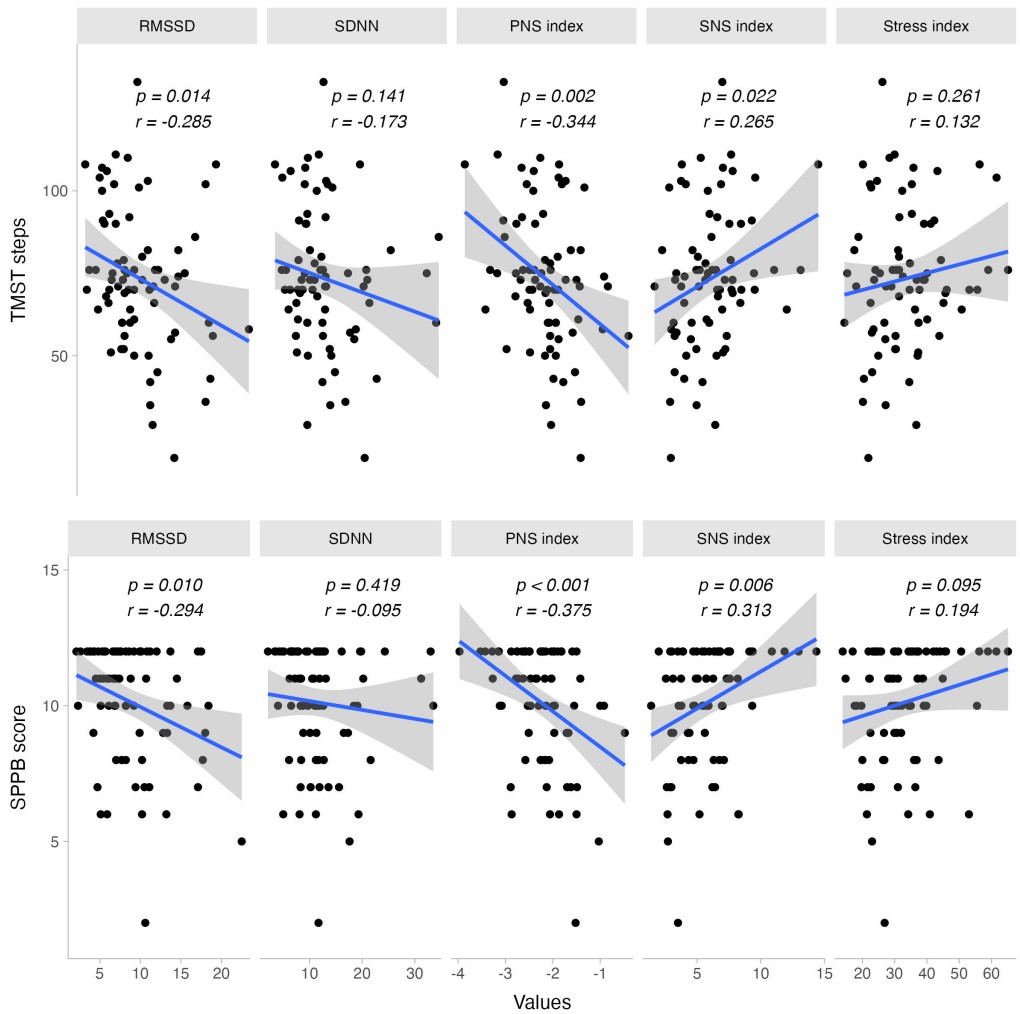

**Figure 4** **Bivariate dispersion plots between physical fitness and HRV-related measures.** Significance values for Pearson's product-moment correlation test are shown.

$CI_{95\%}$ [$-0.111$, $-0.010$], p = 0.014). Similar mediation effects of total muscle mass were observed (ACME = 0.118, $CI_{95\%}$[0.005, 0.350], p = 0.075) when assessing the effect of body fat percentage on TMST steps (ADE = $-0.773$, $CI_{95\%}$[$-1.193$, $-0.300$], $p = 0.001$; Total effect = $-0.655$, $CI_{95\%}$ [$-1.084$, $-0.210$], $p = 0.004$). No other mediating effects were observed for any of the psychological measures or the relationships between body composition and physical fitness measures.

A visual summary of the results is presented in Fig. 5.

### *Post-hoc* power analysis

Finally, when evaluating the statistical power for the observed effects, with correlations of 0.3, 0.4, and 0.5, considering our previously established confidence level and the current sample size, we estimated a statistical power of 76.9%, 95.8%, and 99.8%, respectively.

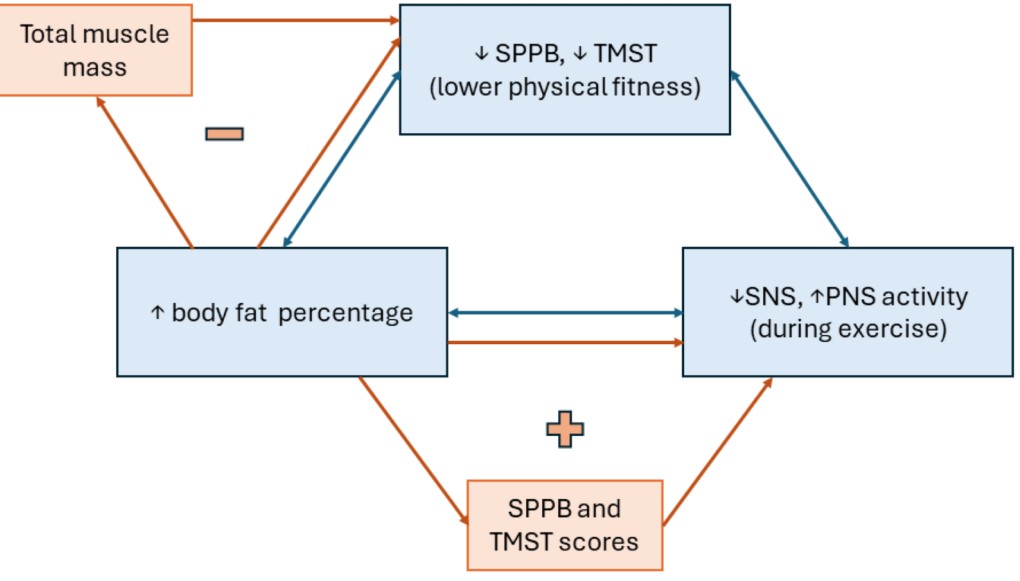

**Figure 5** **Diagram of main results observed from the data.** Blue bidirectional arrows and boxes represent the correlation between variables, while orange unidirectional arrows and boxes represent the factors mediating the above correlations. "+" means that the mediating factor enhances the main observed effect, contrary to "-", reflecting a diminished main effect when considering the mediating factor.

## DISCUSSION

This study investigated the relationship between physical fitness, body fat percentage, and cardiac autonomic response to exercise in community-dwelling older people (aged: 71.5 ± 5.6). We elucidated the interplay between these factors and their implications for cardiac autonomic function during physical efforts through correlational and mediating analyses. Sex-separated analyses were carried out to avoid incorrect interpretation of the results and to clarify the effect of the difference in sample size between men and women on our main results.

A higher body fat percentage was hypothesized to be inversely correlated to physical fitness. Our results align with this hypothesis and previous research regarding body composition and physical performance, with a proportional decrease in older people (*Paranhos Amorim et al., 2022*; *Fatyga-Kotula et al., 2022*). Using the SPPB, we examined three components of physical fitness: gait speed, lower limb relative strength, and balance. Additionally, using the number of steps in the TMST, we assessed aerobic capacity. To better understand the interaction between body composition and physical performance, we analyzed each component separately. We found that the negative correlation with fat percentage and BMI was consistent for gait speed, lower limb relative strength, and aerobic capacity, but not for balance. This analysis contributes to a more specific understanding of how these two widely studied variables are related.

When making this analysis separated by sex, it was found that the main observed correlations maintained significant only for women, which shows that these results were not increased by the presence of a limited number of men in the sample.

However, the mediation analysis showed that total muscle mass diminished the main observed effect of body fat on physical fitness, with more muscled individuals displaying better physical performance despite the body fat percentage. This finding suggests that older adults could benefit from activities promoting muscle mass conservation, even when they have high body fat percentages (*Ramírez-Vélez et al., 2016*), offering insights into the protective role of muscle mass during aging. Nevertheless, caution is advised in interpreting this result, as the close physiological association between increasing body fat and decreasing muscle mass during aging has been well-studied (*Morgan, Smeuninx & Breen, 2020*).

Regarding the link between physical fitness and cardiac autonomic response to exercise, our results evidenced that individuals with a high sympathetic drive while exercising had greater physical fitness and exhibited better performance during TMST. Previous research had demonstrated the utility of HRV and exercise-induced responses in those metrics as a potential marker of cardiorespiratory fitness in these individuals (*Mongin et al., 2022*), supporting the current hypothesis that a greater sympathetic drive facilitates physical performance during moderate-intensity activities. Even more, it has been previously demonstrated that physically active individuals have an enhanced autonomic response to exercise, playing a pivotal role and strengthening this hypothesis (*Sarmento et al., 2017*; *Soares-Miranda et al., 2014*). Despite that, our results provide information about these relationships in individuals aged 60 years or older.

When sex specific influence on main results was analyzed, it was observed that the positive association between sympathetic drive and TMST performance only remained true for women, however, the correlation between sympathetic activity and SPPB sit-to-stand score was stronger in men, showing that men could have had a positive influence in the correlations between physical fitness and autonomic response for the total sample, only in certain variables. This highlights the importance of a more sex-equilibrated sample, as discussed further in the study's lmitations.

Furthermore, the increase in body fat percentage was related to reduced sympathetic activity and a higher parasympathetic activity during TMST, as hypothesized (*i.e.*, SNS index, SI, SDNN, and RMSSD). ANS is expected to shift into a sympathetic predominance throughout physical efforts, ensuring a sufficient response to higher metabolic demands (*Marasingha-Arachchige et al., 2022*). However, aged individuals may present an impaired ANS response to physical exercise (*Castillo-Aguilar et al., 2023*). In that sense, our results support body composition's role in autonomic response. Moreover, they suggest that the negative effects of body fat on physical fitness in older adults may be partially attributable to less efficient autonomic inputs to the cardiovascular system (*Sinha et al., 2023*), adding new insights into this issue. When analyzing this association by sex, it was found that the results remained consistent in sympathetic activity markers form women, but not for men; demonstrating that the observed main effect is not due to men presence. However, the positive correlation between body fat and, RMSSD and SDNN disappeared for both sexes when analyzed separately.

Surprisingly, mediating analyses found that higher physical fitness (SPPB and TMST scores) intensifies the negative impact of body fat on autonomic control. This unexpected finding hints at a complex interplay between physical fitness, adiposity, and cardiac autonomic regulation. Our results indicate that in individuals with greater functional capacity, the influence of body fat on HRV is amplified rather than mitigated. This observation challenges traditional conceptions about the relationship between physical fitness and cardiac autonomic regulation and raises questions about the underlying mechanisms involved (*Tiwari et al., 2021*).

As far as we know, no previous studies have addressed these findings. We posit that further factors exist that influence the mediation effect. For instance, body distribution of the adipose tissue was not explored, especially when it is known that visceral fat has an important impact on cardiovascular health and autonomic regulation (*Triggiani et al., 2019*; *Chait & Hartigh, 2020*). Participants with higher fitness may have a different fat distribution than lower-performance individuals, which may affect the autonomic drive to the heart (*O'Donovan et al., 2009*). Furthermore, other physical, physiological, demographic, or psychological variables may be unexplored. However, these hypotheses must be examined under scientific standards. Therefore, exhaustive research is needed to address this intriguing interplay.

Our results support the interplay between physical fitness, body fat, and cardiac autonomic response to exercise in older individuals. Higher body fat percentage negatively affects physical fitness and is also associated with impaired cardiac autonomic response to exercise, characterized by decreased sympathetic drive and increased parasympathetic tone. Additionally, this altered autonomic response is related to lower physical performance in older individuals, which are important predictors of health outcomes in this population (*Langhammer, Bergland & Rydwik, 2018*; *Tan et al., 2019*). Overall, our findings underscore the triangular and bidirectional relationship among these variables.

However, our results did not evidence a mediating effect of GDS-30 or BAI scores, challenging our initial hypothesis. This finding suggests a nuanced interplay between psychological variables and physical/physiological measures in older people, underscoring the need for further research to elucidate underlying mechanisms and variables beyond depression and anxiety. While the lack of evidence suggesting any mediating effects does not diminish the significance of psychological factors in aging, it highlights the complexity of their influence.

This study was strengthened by the fact that SPPB and TMST measures consistently aligned in their interaction with other variables, which supports a global understanding of physical fitness in older people rather than specific aspects, and by exploring mediating factors in the primary results. Nonetheless, this cross-sectional study is not without limitations. First, our study design prevents us from making causal inferences about the direction of the relationships, highlighting the relevance of experimental and longitudinal research exploring causal relations between our variables and potential underlying mechanisms. Second, a relatively small sample was included and selected with no randomization process, limiting the generalizability and statistical power of the analysis. Furthermore, the sample consisted mainly of women (82.7%), limiting our ability to

conduct sex comparisons. Caution is advised in interpreting sex differences in sample characteristics, presented in Table 1. Given the natural differences in body composition, fitness, and HRV, our findings may be influenced by the presence of 14 males, therefore, we recommend using an equilibrated male/female sample to be able to control sex influences in the future. Further research could investigate a broader group of potentially mediating factors to understand the impact of the personal context and the underlying mechanisms of the associations while using longitudinal designs, which could be useful to improve internal and external validity.

## CONCLUSIONS

The results highlighted a structural bidirectional relationship among variables in our sample. Body fat percentage was inversely correlated to physical fitness and cardiac autonomic activity during exercise, with sympathetic drive associated with physical fitness in older people. Furthermore, physical fitness may mediate the effect of body fat on cardiac autonomic activity during exercise, and total muscle mass may mitigate the negative effect of increased adiposity on physical fitness, highlighting its pivotal role in older people's health. Further research is needed to assess sex influences on this topic.

## ACKNOWLEDGEMENTS

This project was part of Consolidating the binational (Chile-Argentina) research network in Southern Patagonia (FOVI210061).

### Funding

This study was supported with funds from C.N.-E. by ANID Proyecto Fondecyt Iniciación N° 11220116. The funders had no role in study design, data collection and analysis, decision to publish, or preparation of the manuscript.

### Grant Disclosures

The following grant information was disclosed by the authors:
ANID Proyecto Fondecyt Iniciación N°: 11220116.

### Competing Interests

The authors declare there are no competing interests.

### Author Contributions

- Diego Mabe-Castro conceived and designed the experiments, performed the experiments, analyzed the data, prepared figures and/or tables, authored or reviewed drafts of the article, and approved the final draft.
- Matías Castillo-Aguilar performed the experiments, analyzed the data, prepared figures and/or tables, and approved the final draft.

- Matías Mabe-Castro performed the experiments, authored or reviewed drafts of the article, and approved the final draft.
- Ruby Méndez Muñoz conceived and designed the experiments, performed the experiments, authored or reviewed drafts of the article, and approved the final draft.
- Carla Basualto-Alarcón performed the experiments, authored or reviewed drafts of the article, and approved the final draft.
- Cristian Andrés Nuñez-Espinosa conceived and designed the experiments, performed the experiments, analyzed the data, authored or reviewed drafts of the article, and approved the final draft.

## Human Ethics

The following information was supplied relating to ethical approvals (i.e., approving body and any reference numbers):

The Ethics Committee of the University of Magallanes (N° 10/CEC-UMAG/2023) approved this study following the regulations established by the Declaration of Helsinki on ethical principles in human beings.

## Data Availability

The raw data are available in the Supplemental File.

## Supplemental Information

Supplemental information for this article can be found online at http://dx.doi.org/10.7717/peerj.18061#supplemental-information.

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
