# Peer review of "Associations between physical fitness, body composition, and heart rate variability during exercise in older people: exploring mediating factors"

_PeerJ, doi:10.7717/peerj.18061_

## Round 0.1 · original submission · Minor Revisions

Dear Author,

I carefully read the manuscript and the reviewers' comments. I agree with the comments of all the reviewers and ask you to act on them, correct the manuscript according to the comments and resubmit it to the PeerJ journal for consideration.

Kind regards,
Jovan Gardasevic,
Academic Editor

·

Basic reporting

The authors must be commended for carrying out an investigation regarding the relationship between physical fitness, body composition and cardiac autonomic response to exercise in older adults. This topic is important, relevant and relatively novel.
The methodology used in the study is appropriate and it is written with good clarity in professional and unambiguous language.

I appreciate providing the raw data, however the instruments data were missing (SPPB, TMST,…) in the supplemental files.

Discussion
Line 358: I suggest adding a reference at the end of this paragraph, to support the traditional concept and your finding.

Experimental design

No comment

Validity of the findings

No comment

Reviewer 2 ·

Basic reporting

Thank you for the opportunity to review the manuscript: “Associations between physical fitness, body composition, and heart rate variability during exercise in older people: exploring mediating factors”

In general, this is a well-written manuscript on an interesting topic seeking to further understand how body composition is related to physical fitness and HRV. The manuscript has adequate references and the authors did a good job in describing their analyses.

Specific notes
Abstract:
Please remove the word 'sport'. There were no 'sport' applications recorded in this study. There were physical assessments of fitness, but no sport performance measures were listed in the procedures section.

Line 65: Do you mean 'resting' instead of rest?
Line 80: Recommend using the plural 'responses'
Line 322: Please change to 'community dwelling individuals'. Older people is too vague and not helpful in knowing what is considered 'old'.
Line 343: Please change to reflect a more precise age range of the participants

Experimental design

One potential issue with the findings of this study with respect to body composition and physical fitness is the disparity in the equal number of participants for each sex. While pre-adolescent children in fact have very similar body compositions (not much difference in %body fat among young boys and girls), in any population past the age of adolescence, there are much larger differences noted in %body fat among adult men and women. In older adults, this %body fat difference would also be observed. I appreciate the author's attempt to avoid making any sex comparisons in this study, but I am also not convinced that the 67 females are enough to fully understand this relationship between %body fat and physical fitness, because the 14 males had significantly greater muscle mass and significantly lower fat mass compared to the females in the study, which seemed to affect the Overall means by 2-3 kg of muscle mass and Fat mass. So if 14 males changed the Overall mean by that much, what would 67 males do? Because the overall analyses were performed in a combined manner (Overall mean, as opposed to separation by sex), I feel this is a big limitation when interpreting these findings.

Line 123: The power analysis performed is correct, however there were only 81 included in the study. I appreciate the flow chart explaining how 100 participants were originally recruited, and due to cognitive factors, and dropout, only 81 were included; yet 82 were explicitly needed according to G*Power in order to protect against a type II error with 80% power. I am not convinced that one additional participant would ultimately change the main results of this study, however, 82 were required, and there were not 82 participants.

Validity of the findings

This is a meaningful topic, and the statistical methods are sound given the methods the authors chose. In the bivariate dispersion plots, it would be helpful to also include the r-values along with the p-values for the reader to see both in one place.

The conclusions were well stated, given the dataset, and I do believe this is an important topic we need to further understand. However, community dwelling individuals are not a 'difficult' population to study (as opposed to say children where parental consent is required, or in other protected populations). My recommendation for the authors is to find another community dwelling location to collect additional data from. Then they can either find enough women in the study to be above 82, and keep the results limited to women, or attempt to get men and women up above 64 in each group (G*Power requires 64 in each group when comparing independent samples t-tests, two-tailed, p = 0.05, beta = .80, with a moderate effect size = 0.5) and make sex comparisons to determine if the %body fat associations are different among men and women of similar ages.

Additional comments

I suggest the authors recruit additional participants from another community dwelling center to increase the size of their sample, or to conduct post-hoc power analyses using the 67 females participants only. I feel the 14 male participants may have influenced the overall means by quite a lot (only 14 males created a change in 2-3 kg in muscle mass and fat mass, so how much would 30 or 60 participants change the mean)?

Reviewer 3 ·

Basic reporting

The present manuscript is clear in writing and has important notes, results, and interpretations of the data.
Figures and tables are legible.
The introduction can use a little more highlight on the relevance of HRV and the operational definitions of the variables. Which one is most important to report? How can someone measure their own HRV?

Experimental design

The methods and experimental design are well-developed. The only challenge I will provide is that I do not think it is necessary to conduct Pearson correlations between the SPPB score and body fat. An alternative suggestion would be to complete the correlation on the raw test scores that comprise of the SPPB, separately.

Validity of the findings

Considering the last comment, I would suggest to remove the presented correlations with the SPPB score (and respective figures).
In Table 1, the sample characteristics are presented and include 95%CI. I think it is important to denote or highlight where sex differences exist, because when the CI includes "0" (zero), then there is no difference in means, but if the CI does not contain 0, there there is a difference. This is the case, and expected, for height, weight, muscle mass, fat mass, bone mass, and % water.
Further, beyond these sex differences, none of the data should be presented with sex differences because of the substantial differences in sample size between the sexes.

Additional comments

Figure 5 is a great addition to the literature. I appreciate its simplicity yet ability to convey the complexity of the interrelationships among the variables.
Lines 90-93 is redundant with lines 77-84. Please omit lines 90-93.
Line 332: Please omit "new perspective" from this line. There is a substantial body of literature that presents the importance of muscle mass and aging. This is not a new concept.

---

## Round 0.2 · Minor Revisions

Dear author,

Reviewer 3 requires minor revisions to the manuscript to make it ready for publication in this respected journal. Make the requested changes and resubmit the manuscript.

Kind regards,
Academic Editor

Reviewer 2 ·

Basic reporting

Thank you for making the appropriate changes to the manuscript.

Experimental design

Thank you for describing the complexities of the sample size and how further subjects are not able to be included at this point.

Validity of the findings

I appreciate the authors' attempt to perform the post-hoc power analysis to better validate the findings.

Additional comments

These changes have improved the manuscript.

Reviewer 3 ·

Basic reporting

no comment

Experimental design

The methods are mostly clear and can be replicated.

Validity of the findings

Again, there are some questions that were not originally addressed. These include the sample differences in males and females (thank you for adding a limitation statement in the Discussion).

Additional comments

-With the disparity of body composition variables between sex, it may be in the best interest of the manuscript to eliminate the male sample, as it is very disproportionate to the sample of females. This will alter the results but will be more reflective of the actual relationships with the other dependent variables considering most regression models include sex as separate variables anyway, not combined. If keeping the males in, please run separate correlations on an only female sample and add to the manuscript to identify exactly how males influence the results.
The alternative (as mentioned by other reviewers) would be to collect more male data.
-Lines 265-271. The correlations for each individual SPPB assessment results (i.e., time of 4-meter walk, time of repeated chair rise, etc.) should be reported. If not eliminating the correlation with the SPPB composite score (which does not statistically make sense because it is an ordinal value), the individual test scores should be correlated for a more global understanding of each component of the SPPB and how they each correlate to body comp-related variables, rather than the composite SPPB score.

---

## Round 0.3 · accepted · Accept

Dear author,
after revising the article in relation to the reviewers' comments, the manuscript is now, in my opinion, ready for publication.

Kind regards,
Academic Editor

Reviewer 2 ·

Basic reporting

n/a

Experimental design

n/a

Validity of the findings

Separating some of the statistical findings according to sex definitely helps add transparency to this observation. Yes there is a limitation in the sample size, but that is now clearly defined in the limitations section.

Additional comments

n/a

Reviewer 3 ·

Basic reporting

No additional comment

Experimental design

No additional comment

Validity of the findings

No additional comment

Additional comments

Thank you for making the requested modifications.